# Tumor-infiltrating Leukocytes Suppress Local Inflammation Via Interleukin-1 Receptor Antagonist in a Syngeneic Prostate Cancer Model

**DOI:** 10.3390/biology9040067

**Published:** 2020-03-31

**Authors:** Yu-Ching Fan, Wei-Yu Chen, Kuan-Der Lee, Yuan-Chin Tsai

**Affiliations:** 1Ph.D. Program for Cancer Molecular Biology and Drug Discovery, College of Medical Science and Technology, Taipei Medical University, Taipei 11042, Taiwan; ssfan1616@gmail.com; 2Department of Pathology, Wan Fang Hospital, Taipei Medical University, Taipei 11042, Taiwan; 1047@tmu.edu.tw; 3Department of Pathology, School of Medicine, College of Medicine, Taipei Medical University, Taipei 11042, Taiwan; 4Department of Hematology and Oncology, Taipei Medical University Hospital and Department of Medicine, Taipei Medical University, Taipei 11042, Taiwan; kdlee@h.tmu.edu.tw; 5Graduate Institute of Cancer Biology and Drug Discovery, College of Medical Science and Technology, Taipei Medical University, Taipei 11042, Taiwan

**Keywords:** tumor microenvironment, tumor-infiltrating leukocytes, interleukin-1 receptor antagonist

## Abstract

Background: Several lines of evidence have demonstrated the tumor-promoting function of inflammation. Since many chemokines are important in coordinating immune cells during inflammation, monitoring intratumoral chemokines provides a way to study the tumor microenvironment. Methods: To identify tumorigenic chemokines, we compared two syngeneic mouse prostate cancer cell lines by an antibody array and quantitative reverse-transcription polymerase chain reaction (RT-PCR). The tumor microenvironment was analyzed by monitoring gene expressions in mouse tumor tissues, primary cells, and tumor-infiltrating leukocytes (TILs). Result: We identified a group of pro-inflammatory chemokines associated with a tumorigenic transgenic adenocarcinoma mouse prostate (TRAMP)-C1 cell line. In the tumor microenvironment, the TILs secrete a natural anti-inflammatory factor, interleukin-1 receptor antagonist (IL1RN), which inhibits the functions of pro-inflammatory molecules and likely accounts for tumor type-specific anti-inflammation functions. Conclusion: Our results support that tumor cells recruit TILs by pro-inflammatory chemokines to establish an IL1RN-mediated anti-inflammatory environment in the syngeneic prostate cancer model.

## 1. Introduction

Tumor-promoting inflammation has been recognized as one important feature during tumorigenesis [1]. In the mouse skin carcinogenesis model, mice with deletion of tumor necrosis factor α (TNFα), a master pro-inflammatory cytokine, were refractory to development of either benign or malignant tumors, demonstrating the tumor-promoting role of inflammation [2]. In prostate cancer, many inflammatory leukocytes are recruited into the tumor microenvironment after androgen-deprivation therapy (ADT) [3,4,5]. Their dynamic interactions with tumor cells facilitate development of castration-resistant prostate cancer (CRPC) [4,5]. In addition to TNFα, other cytokines (e.g., interleukin-1, IL1) stimulate an inflammatory program and coordinate tumor-infiltrating leukocytes (TILs) in tumor milieu via activation of pro-inflammatory chemokines [6]. Chemokines are categorized into four subfamilies based on the configuration of the N-terminal cysteine residues such as that the C-C group has two sequential cysteines (e.g., CCL2) and the C-X-C subfamily has an amino acid inserted in the middle (e.g., CXCL1 and CXCL10) [7]. Several chemokines are also involved in tumor-promoting function. For example, similar to TNFα deletion, CCL2 knockout mice also exhibit resistance to skin tumor formation [2].

The interleukine-1 receptor antagonist (IL1RN, also known as IL-1Ra) belongs to the IL1 family and competes with other IL1 ligands (e.g., IL1α and ILβ) to IL1 receptor; however, due to lack of interaction with the co-receptor, interleukin-1 receptor accessory protein, IL1RN does not lead to stimulation of inflammatory signaling [8]. Therefore, IL1RN is a natural anti-inflammatory cytokine, and using recombinant IL1RN protein has shown therapeutic benefits in a plethora of inflammation-related disease models [9]. The role of IL1RN in tumor biology remains controversial. Several studies have shown that IL1RN can act as an anti-tumor agent partly through inhibiting the IL1 signaling [10,11]. However, it was reported that IL1RN secreted by macrophage can promote growth of normal prostate epithelial cells [12]. The distinct effects may reflect the complicated roles of inflammation in tumor microenvironment.

We hypothesized that tumor chemokines affect the content of tumor microenvironment and tumor progression. We utilized a syngeneic tumor model using the prostate cancer cell lines derived from the transgenic adenocarcinoma mouse prostate (TRAMP) model [13]. After comparing chemokine profiles between the tumorigenic TRAMP-C1 and the non-tumorigenic TRAMP-C3 mouse cell lines [13], we identified a group of pro-inflammatory chemokines in TRAMP-C1. Unexpectedly, using the TRAMP-C1 cell line in a syngeneic tumor model, the chemokines were reduced in the in vivo-derived tumor tissues compared with their corresponding primary cells prepared in vitro. In contrast, those chemokines from the mouse Lewis lung carcinoma cells (LLC) derived tumors showed elevated expressions. Therefore, these results suggest a feature of tumor type-specific microenvironment in that different tumor types lead to distinct inflammatory outcomes in the tumor milieu. We demonstrate that tumor-infiltrating leukocytes (TILs) secreted IL1RN in TRAMP-C1-derived tumor microenvironment and that IL1RN can inhibit the functions of pro-inflammatory cytokines in vitro. Based on these results, we propose a model that some tumor cells, like TRAMP-C1, suppress the inflammatory signaling in tumor microenvironment through the inhibitory effects of IL1RN.

## 2. Materials and Methods

### 2.1. Preparation of Primary Cells and Tumor-Infiltrating Leukocytes (TILs)

The tumorigenic TRAMP-C1, the non-tumorigenic TRAMP-C3, and Lewis lung carcinoma (LLC) cell lines were purchased from the American Type Culture Collection (ATCC; Manassas, VA, USA). All cells were maintained in the suggested culture medium according to the ATCC. Animal experiments were performed in accordance with a protocol approved by the Taipei Medical University Animal Care and Use Committee (number: LAC-2017-0274, Taipei, Taiwan). Two sequential rounds of subcutaneous (s.c.) injections of tumor cells into the right flank of eight-week-old male C57BL/6 mice (NLAC, Taipei, Taiwan) were performed. In the first set of experiments, 1 × 10^6^ TRAMP-C1 cells were injected into two mice, and in the second set, the primary cells derived from the first set of experiment were inoculated into three mice. When performing the same experiment using LLC cell line, we reduced the cell numbers to 2 × 10^5^ cells. After the tumors reached a size around 1000 mm^3^ (length × width^2^ × 1/2), they were dissected and incubated with collagenase type IV (10 ng/mL, Gibco, Grand Island, NY, USA) and DNase type IV (200 U/mL, Sigma-Aldrich, St. Louis, MO, USA), then were filtered through a nylon mesh with 100-µm pore size (Falcon Cell Strainers, Thermo Fisher Scientific, MA, USA) to obtain single-cell suspensions. The cells were re-suspended in phosphate-buffered saline (PBS) and carefully layered onto a Ficoll solution (Ficoll-Paque Plus density gradient media, GE Healthcare Life Sciences, Sweden) followed by a discontinuous density gradient (5 mL PBS plus 3 mL Ficoll) with a centrifugation condition 3000× *g*, 30 min. Under this condition, we validated that the peripheral blood mononuclear cells (PBMCs) would locate in the interphase (Appendix A, left panel) and that TRAMP-C1 cells would be pelleted down in the tube (Appendix A, right panel). Thus, to prepare “primary cells”, which consisted of most TRAMP-C1 cells in the tumor tissues, cells collected from the bottom of the Ficoll gradient were cultured in Petri dishes and passaged for three times before further applications (e.g., preparation of messenger (m)RNA and conditioned media). To prepare tumor-infiltrating leukocytes (TILs), the cell population in the interphase of the Ficoll gradient was collected. Although the Ficoll density procedure is usually used to prepare lymphocytes and monocytes, it was shown that neutrophils could also be enriched [14]. Considering that the procedure can enrich mixed populations; therefore, we collectively called the cell population tumor-infiltrating leukocytes (TILs).

### 2.2. Mouse Chemokine and Cytokine Array Analysis

The manufacturer’s guidelines were followed for this analytical procedure. In brief, conditioned medium was collected from cell lines, primary cells and TILs. Samples were incubated overnight with a membrane containing 40 antibodies (Proteome ProfilerTM Array, R&D System, Minneapolis, MN, USA).

### 2.3. Quantitative Reverse-Transcription (RT)-Polymerase Chain Reaction (qPCR)

Total RNA was isolated from different types of samples followed by TRIzol (Invitrogen, Carlsbad, CA. USA) and RNA isolation system (QIAGEN, Venlo, the Netherlands). For complimentary (c)DNA preparation, 1 µg of total RNA was used for RT (Invitrogen). The amplification step was performed and measured with SYBR Green PCR master mix (Applied Biosystems, Beverly, MA, USA). For all primer pairs, the thermocycler was run for an initial 95 °C for 10 min, followed by 40 cycles of 95 °C for 15 s and 60 °C for 1 min. All reactions were normalized to mouse *Gapdh* and analyzed in triplicate. All primers used are listed in Appendix A.

### 2.4. Immunohistochemistry (IHC)

IHC was performed according to the staining protocol of the UltraVision Quanto Detection System HRP kit (Thermo Fisher Scientific, MA, USA). The CD68 primary antibody (Abcam, Cambridge, United Kingdom) was used followed by procedures of visualization. After counterstaining with hematoxylin, images were analyzed under a microscope.

### 2.5. Statistical Analysis

All data are presented as the mean ± standard deviation (SD). Differences between individual groups were analyzed by Student’s *t*-test.

## 3. Results

### 3.1. Association of Pro-Inflammatory Chemokines and the Tumorigenic Prostate Cancer TRAMP-C1 Cell Line

Our earlier studies showed that human chemokine CCL2 contributes to malignant progression of prostate cancer (e.g., proliferation and invasion) [15]. Since CCL2 is important for recruiting macrophage [16], we questioned whether other chemokines or cytokines are involved in tumor progression. To address this issue, we chose the well-established syngeneic cell lines derived from the transgenic adenocarcinoma mouse prostate (TRAMP), and compared conditioned medium derived from the tumorigenic TRAMP-C1 and non-tumorigenic TRAMP-C3 cell lines [13]. We found increased protein levels of three chemokines (CXCL10, CXCL1, and CCL2) in TRAMP-C1 (Figure 1A, arrows). The conclusion was also confirmed by comparing mRNA levels between the two cell lines (Figure 1B). Since the three chemokines belong to pro-inflammatory factors [7], we confirmed that they were inducible by inflammatory signals such as lipopolysaccharide (LPS) or TNFα (Figure 1C). However, receptors (CCR2 and CCR4) responsible for recognizing CCL2 were suppressed by the same signals (Figure 1D). In summary, we identified a group of pro-inflammatory chemokines that was associated with the tumorigenic TRAMP-C1.

### 3.2. Tumor Type-Specific Anti-Inflammation in TRAMP-C1-Derived Tumor Microenvironment

The association between the pro-inflammatory chemokines and the tumorigenic TRAMP-C1 may suggest an inflammatory process during tumor progression. To examine this hypothesis, we monitored chemokine expressions in tumor tissues collected from two immune-competent mice and their derived “primary cells”, which consisted of most TRAMP-C1 cells in the tumor tissues (Figure 2A). Unexpectedly, after comparing mouse *Ccl2* levels between tumor tissues and their derived primary cells, we found reduced, not increased, expressions in tumor tissues (Figure 2A). Although CCL2 receptor (*Ccr2*) was suppressed in vitro by inflammatory signals (Figure 1D), it was elevated in tumor tissues (Figure 2B). Thus, expression patterns of *Ccl2* and *Ccr2* indicated that inflammatory signals were inhibited in the TRAMP-C1-derived tumor microenvironment. Consistently, we observed reduced expression of *Rela*, one component of the nuclear factor (NF)-κB transcription factor that mediates downstream responses of inflammation signals [7], in tumor tissues relative to primary cells (Figure 2C). To further address whether this inhibitory environment to inflammation might be specific to TRAMP-C1 cell line-derived tumor tissues, we also monitored the expression of the same genes in the Lewis lung cancer cell line (LLC). Different from TRAMP-C1, it was reported that the tumor microenvironment of LLC promoted *Ccl2* expression [17]. Indeed, we confirmed elevated *Ccl2* in a LLC-derived tumor tissue compared with its derived primary cells (Figure 2D). Since CCR2 is crucial for many types of leukocytes to be recruited into tumor microenvironment via CCL2, the high expression of *Ccr2* mRNA observed in both TRAMP-C1 and LLC derived tumor tissues may reflect the enrichment of leukocytes. However, it also suggests distinct inflammatory responses between the two cell lines originated from different tumor types. In summary, our results support a tumor type-specific anti-inflammation in TRAMP-C1 derived tumor microenvironment.

### 3.3. Examination of Tumor-Infiltrating Leukocytes (TILs) in Tumor Microenvironment

In the collected tumors, we observed capillary-like structures encircling red blood cells (small arrow), lymphocytes (middle arrow) and leukocytes with polymorphonuclear feature (big arrow), consistent with the infiltration of immune cells in tumor through angiogenesis (Appendix A). If the recruited immune cells did not express selected chemokines, then that could partly account for reduced levels of pro-inflammatory chemokines in the TRAMP-C1-derived tumor tissues. To examine this possibility, we repeated the s.c. procedures in three mice and prepared the tumor tissues (tumor #1-#3) and their corresponding TILs (TILs #1-#3, Figure 3A). We used the primary cells from TRAMP-C1 cells-derived tumors (Figure 2A) to examine whether the inhibited inflammation can be observed again in the tumor microenvronment. Indeed, we demonstrated anti-inflammatory status as evidenced by reduced *Ccl2* (Figure 3B) and *Cxcl1* (Figure 3C) in all the tumor tissues compared to the primary cells. We were able to detect macrophages by an anti-CD68 antibody in the tumor tissues (left panel, Figure 3D). In isolated TILs, we also observed enriched macrophages (large arrow) and lymphocytes (small arrow) (right panel, Figure 3D). Using lymphocyte markers, we confirmed the enrichment of T cells (*Cd3e*) (2-8 fold, Figure 3E) and B cells (~4 fold, Appendix A, left panel) in TILs relative to tumor tissues. In addition, we also found the subtypes of T cells (CD4 and CD8) were enriched in TILs compared to tumor samples (Appendix A, middle and right panels). Based on these results, we demonstrated that the TILs contained multiple types of leukocytes (e.g., granulocyte, lymphocytes, monocyte/macrophage) in the tumor microenvironment. Furthermore, we also monitored two characteristic features of macrophages subtypes showing opposite roles in inflammation and tumor [18]. We monitored M1 subtype by measuring nitric oxide synthases 2 (*Nos2*, Figure 3F) and M2 class with arginase 1 (*Arg1*, Figure 3G). In summary, we demonstrated the presence of TILs in the tumor microenvironment.

### 3.4. Expression and Secretion of Interleukin-1 Receptor antagonist (IL1RN) in Tumor-Infiltrating Leukocytes (TILs)

We hypothesized that TILs can modulate the inflammatory process in tumor microenvironment; therefore, we sought to monitor protein secretion profile by the TILs. We compared the conditioned mediums derived from peripheral blood mononuclear cells (PBMC) of healthy mice and those from TILs. As shown in Figure 4A, we found that all the conditioned mediums derived from TILs (TILs #1-#3) showed an increased abundance of IL1RN (dashed box) compared with PBMC (PBMC#1, PBMC#2). When monitoring the mRNA level of *Il1rn,* tumor tissues collected from mice showed increased levels compared with the derived primary cells in vitro (2–6 fold, Figure 4B). This pattern was distinct from the reduced expressions of *Ccl2* and *Cxcl1* in tumor tissues compared with primary cells (Figure 3B,C). Although the tumor tissues contain a mixed population of cells (e.g., prostate cancer cells and immune cells), when comparing tumor tissues with their corresponding TILs, TILs showed increased expression of *Il1rn* (6–12 fold, Figure 4C). Therefore, we concluded that TILs, not cancer cells, were the primary source for expressing and secreting IL1RN in the tumor microenvironment in the syngeneic prostate cancer model.

Earlier studies have found that both monocytes and polymorphonuclear cells from human circulating peripheral blood can express IL1RN [19]. In addition, cell culture experiments showed that using human monocyte cell line THP-1 and also its differentiated macrophage can induce IL1RN expression after direct contact with T cells [20]. Therefore, it is possible that the monocytes and granulocytes could be responsible for the IL1RN expression in the TILs. The complex interactions with T cells and other types of cells, as detected in our enriched TILs (Figure 3), may play an important role for IL1RN expression in the tumor microenvironment.

### 3.5. Inhibitory Effects of IL1RN on Pro-Inflammatory Cytokines

IL1RN is a natural anti-inflammatory factor that binds to the IL1 receptor without transducing intracellular signaling, thus competing for the binding of the pro-inflammatory cytokine IL1β [9]. When TRAMP-C1 cells treated with IL1β at a concentration as low as 0.1 ng/mL induced efficient up-regulation of pro-inflammatory chemokines (Ccl2, Cxcl1, Cxcl10), exogenous IL1RN inhibited IL1β function in a concentration-dependent manner (Figure 5A). It was shown that IL1RN has beneficial effects in treating rheumatoid arthritis where the pro-inflammatory cytokine TNFα is a driver for the pathological progression [9]. It is possible that IL1RN may compromise the downstream effects of TNFα. Indeed, IL1RN was capable of inhibiting chemokine induction in TRAMP-C1 cells in response to lower dosage of TNFα treatment (Figure 5B). We also examined whether IL1RN can be regulated by pro-inflammatory signals and found that TRAMP-C1 cell line treated with either LPS or TNFα did not affect IL1RN expression (Figure 5C), supporting that the anti-inflammatory effects in tumor tissues are due to those recruited TILs, but not TRAMP-C1 cells. In summary, our results support a working model that tumor cells can recruit TILs through tumor-derived chemokines, and TILs suppress inflammation in the local environment via IL1RN-mediated functions (Figure 5D).

## 4. Discussion

We identified several pro-inflammatory chemokines that were intrinsically expressed in the tumorigenic TRAMP-C1 cell line but were suppressed in its derived tumor milieu. The underlying mechanism is due to anti-inflammatory functions of IL1RN secreted by TILs. As a comparison, we also analyzed the mRNA samples from LLC-derived tumor, which was suggested to be under inflammatory stress. Although the pro-inflammatory chemokine *Ccl2* was highly activated in the LLC-derived tumor (Figure 2E), we found that the *Il1rn* level was increased in LLC-derived tumor tissue (~four-fold) and its corresponding TILs (~50-fold) when they were compared to LLC cell line (Appendix A, left panel). However, the *Cxcl1* level, another pro-inflammatory chemokine, was drastically increased (~30–60-fold) in the same LLC-derived tumor samples, which is the opposite to the suppressed features (~7–8-fold) shown in TRAMP-C1 derived tumor tissues (Figure 3C). These results are consistent with the tumor type-specific microenvironment in that LLC-derived tumors exhibit an inflammatory status while TRAMP-C1-derived ones show anti-inflammation. Since tumor type-specific microenvironments might be determined by multiple mechanisms, it is possible that LLC cells stimulate stroma and immune cells to overcome the anti-inflammatory effects of IL1RN. Supporting this hypothesis, it was reported that macrophages release TNFα, which is crucial for the elevated CCL2 expression in the LLC-derived tumors [17]. Therefore, increasing TNFα abundance should eventually override the function of IL1RN in vivo, which is consistent with our results that cells treated with higher TNFα (1 ng/mL) were insensitive to the inhibitory effect of exogenous IL1RN (Figure 5B). This provides one explanation how LLC-derived tumors still exhibit inflammatory features while having high levels of IL1RN.

It is likely that TILs are recruited to tumor tissues in part through pro-inflammatory chemokines (e.g., CCL2) and local angiogenesis. The TRAMP-C1-secreted chemokines, especially CCL2, are also known to be responsible for the initial signals for promoting angiogenesis [21]. Based on our proposed model, it depicts the sequential events in establishing an anti-inflammatory microenvironment (Figure 5D). However, the consequence of this setting remains unclear. Based on earlier studies using immunohistochemical analysis, IL1RN seems to have lower expression in prostate cancer samples compared with normal tissues [22]. We also analyzed the mRNA levels of IL1RN in different tumor stages from a published clinical dataset [23]. Although the samples showed broad distribution and no statistical significance, the average expression values from tumors with malignant stages (pT3a/pT3b: 0.924/1.02) were slightly lower than earlier stage (pT2c: 1.039) (Appendix A, left panel). Therefore, these results and others [10,11], support an anti-tumor role of IL1RN, which maybe be involved in inhibiting tumor-promoting inflammation. The negative association between IL1RN and tumor progression in the earlier study (ref. [22]) recapitulates molecular features of the immunoediting theory, which consists of three major phases of elimination, equilibrium, and escape [24]. It is possible that that the initial immune-activation function of increased inflammation after IL1RN reduction does cause tumor cell death in the elimination phase, but it also accelerates the process of selection for resistance to immune surveillance functions. Thus, similar to the tumor-promoting effect of inflammation as demonstrated by TNFα in skin carcinoma model [2], tumor cells may eventually escape the immune system by reducing the IL1RN level in the tumor microenvironment and develop into malignant cancer in late stages.

It is known that the androgen receptor (AR) is a key driver of prostate cancer and inhibition of AR function by ADT is the mainstay of the current therapeutic strategy [25,26,27]. After ADT, many kinds of leukocytes were shown to be recruited into the tumor microenvironment [3,4,5]. Following the idea of anti-cancer function of IL1RN [10,11], we further compared the mRNA levels of IL1RN before ADT (pre-ADT) and after ADT (post-ADT, 22 weeks after ADT) in seven prostate cancer patients [28]. As shown in Appendix A (right panel), we found increased average level of IL1RN in patients after ADT. It is known that patients treated with ADT usually develop CRPC after 2-3 years; therefore, it is possible that IL1RN serves to slow down the progression rate. It should be noted that most ADT samples from the study [28] were derived from patients treated with ADT for less than one year. It would be interesting to monitor whether the IL1RN levels are indeed lower in CRPC patients compared to those in ADT patient. Since tumor cells death due to androgen deprivation already imposes strong inflammation stress in the tumor microenvironment, increased IL1RN may be a natural host response to ameliorate the collateral tissue damages, which also serves to delay the malignant progression. However, we are aware that IL1RN and its related TILs may also serve as tumor-promoting agents facilitating tumor progression in certain conditions. It was reported that using a 3D culture system, IL1RN can promote cell proliferation of normal prostate epithelial cells [12]. The IL1RN among other factors is partly responsible for the growth-promoting effect of a macrophage cell line, Raw 264.7 [12]. It is possible that in combination with other cytokines/chemokines IL1RN did trigger different types of signaling programs facilitating proliferation and malignant progression. In tumorigenesis, the cellular contents in different stages may serve a determining role of different responses to IL1RN. Most prostate cancers have *Pten* and *Trp53* defects, it is unclear whether perturbation of the cellular signaling inside cells leads to different cellular responses compared to normal epithelial cells. Further investigation is warranted to understand the role of IL1RN in tumor biology.

## 5. Conclusions

We found that tumor-infiltrating leukocytes are crucial for establishing a tumor type-specific anti-inflammatory microenvironment partly through secretion of IL1RN.

## Figures and Tables

**Figure 1 biology-09-00067-f001:**
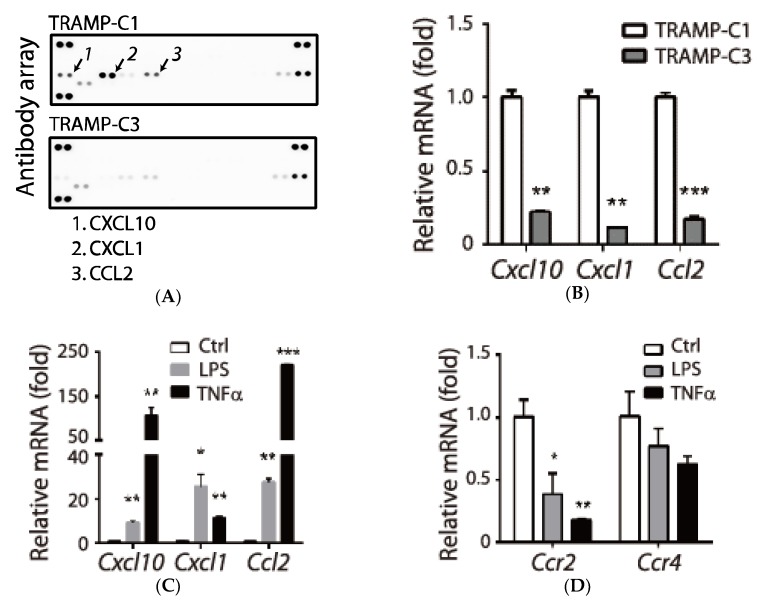
Association of pro-inflammatory chemokines with the tumorigenic prostate cancer cell line transgenic adenocarcinoma mouse prostate (TRAMP)-C1. (**A**) Identification of tumor-associated chemokines (arrows) by comparing conditioned media prepared from TRAMP-C1 and non-tumorigenic TRAMP-C3 cells using an antibody array. Reference spots at the corners, except for the bottom-right one, served as quality control. Induced genes in TRAMP-C1 are labeled with numbers. (**B**) Relative mRNA levels between TRAMP-C1 and TRAMP-C3 cells were determined by a real-time qPCR. (**C**) Induction of tumor-associated cytokines by inflammatory signals. TRAMP-C1 cells were treated with lipopolysaccharide (LPS, 100 ng/mL) or mouse tumor necrosis factor α (TNFα, 10 ng/mL) for 6 h followed by mRNA purification and a real-time qPCR analysis. (**D**) Suppression of *Ccr2* expression by inflammatory signals. Samples from panel C were analyzed using primers against *Ccr2* and *Ccr4* by a qPCR. Control: vehicle treatment. qPCR measurements were derived from three technical replicates and results are presented as the mean ± SD. Student’s *t*-test. * *p* < 0.05, ** *p* < 0.01, *** *p* < 0.001.

**Figure 2 biology-09-00067-f002:**
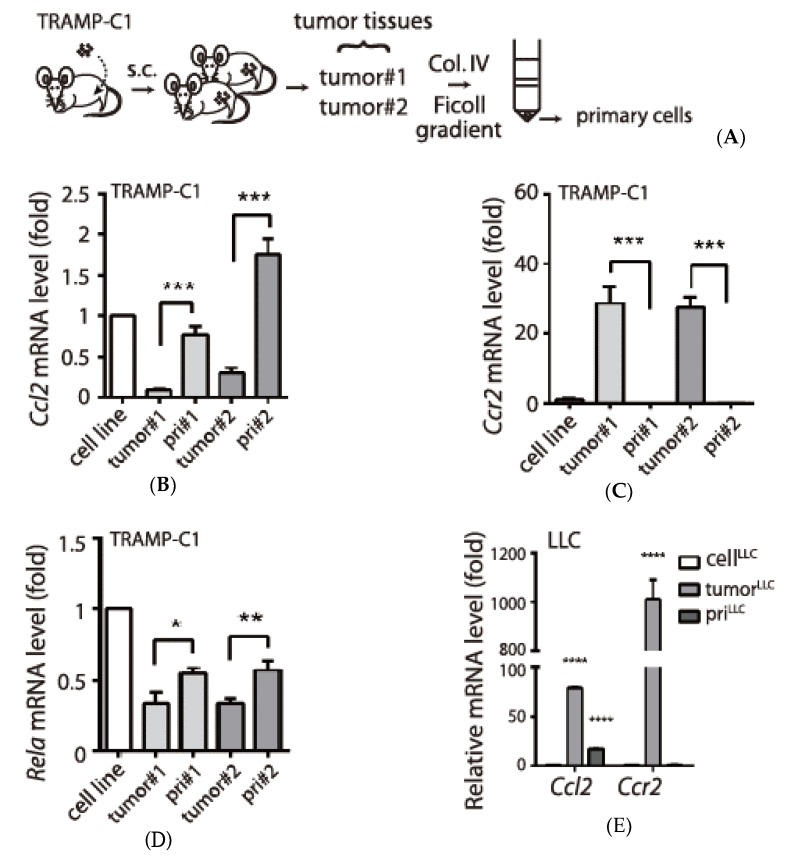
TRAMP-C1 derived tumor microenvironment shows tumor type-specific suppression of inflammatory signaling. (**A**) The procedure for obtaining two independent tumor tissues (tumor#1-#2) and primary cells. The tumor tissues were treated with collagenase/DNase treatment (Col. IV) followed by Ficoll gradient. (**B**) Reduction of *Ccl2* mRNA in the TRAMP-C1 cell line-derived tumor environment. *Ccl2* mRNA levels were compared in the syngeneic TRAMP-C1 cell line, tumor tissues (tumor#1 and tumor#2), and their derived primary cells (pri#1 and pri#2). (**C**) Increased expression of *Ccr2* mRNA in the TRAMP-C1-derived tumor environment. A real-time qPCR was performed using the same samples in panel A. (**D**) Reduction of *Rela* mRNA in the TRAMP-C1-derived tumor environment. (**E**) Lewis lung cancer (LLC)-derived tumor environment has elevated *Ccl2* expression. mRNA levels were compared in the LLC cell line (cell^LLC^), its tumor tissue (tumor^LLC^), and its derived primary cells (pri^LLC^). qPCR measurements were derived from three technical replicates and results are presented as the mean ± SD. Student’s *t*-test. * *p* < 0.05, ** *p* < 0.01, *** *p* < 0.001, **** *p* < 0.0001.

**Figure 3 biology-09-00067-f003:**
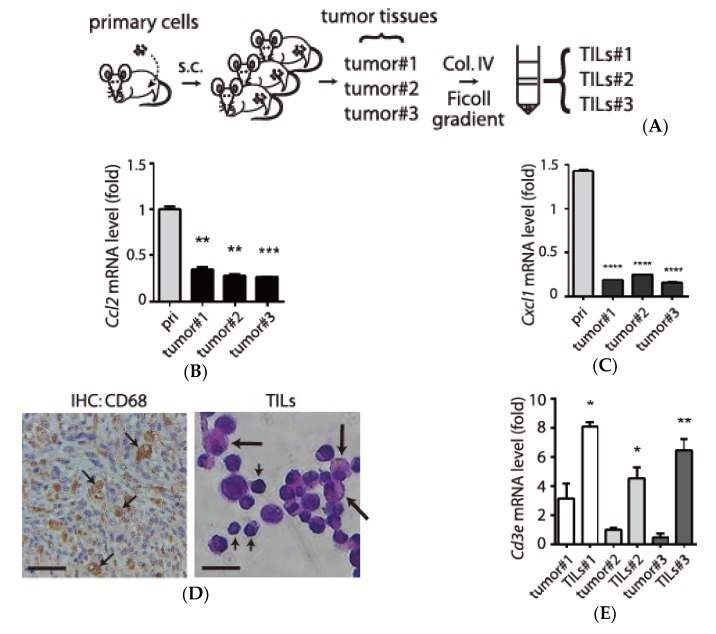
Isolation and characterization of tumor-infiltrating leukocytes (TILs) in the tumor microenvironment. (**A**) The schematic drawing for obtaining three independent prostate tumor tissues (tumor#1-#3) and their corresponding TILs (TILs #1-#3). Primary cells prepared from TRAMP-C1 cell line-derived tumors were subcutaneously (s.c.) injected into three C57BL/6 mice. The tumor tissues were treated with collagenase/DNase treatment (Col. IV) followed by Ficoll gradient to enrich TILs. (**B**,**C**) Validation of reduced inflammatory chemokines mRNA in tumor tissues (*Ccl2*, panel B) and (*Cxcl1*, panel C). mRNA levels of tumor tissues (P2-T, P3-T1, P3-T2, and P3-T3) were normalized to *Gapd*h for comparison with primary cells (pri.). (**D**) Left: selected images of macrophages in tumor tissues (arrows) by immunohistochemistry (IHC) using an antibody against CD68. Scale bar: 10 µm. Right: isolated TILs visualized with Hematoxylin and Eosin (H&E) staining. Large arrows: macrophage, small arrows: lymphocytes. Scale bar: 20 µm. (**E**) Enrichment of lymphocytes in the tumor tissues and TILs by monitoring mRNA levels of a lymphocyte marker *Cd3e.* (**F**,**G**) Monitoring subclasses (M1 and M2) of macrophage in the TILs by measuring an M1 marker *Nos2* (panel F) and an M2 marker *Arg1* genes (panel G). qPCR measurements were derived from three replicates and results are presented as the mean ± SD. Student’s *t*-test. * *p* < 0.05, ** *p* < 0.01, *** *p* < 0.001, **** *p* < 0.0001.

**Figure 4 biology-09-00067-f004:**
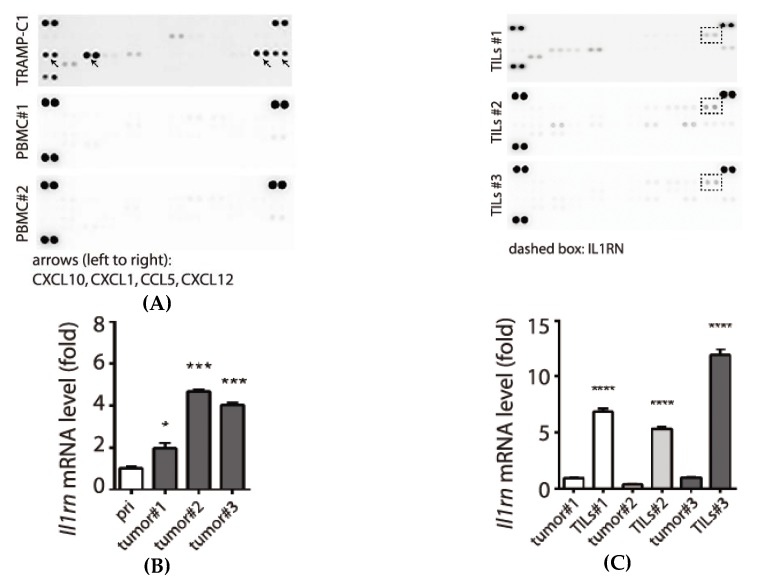
Secretion of interleukin-1 (IL1) receptor antagonist (IL1RN) by TILs. (**A**) Analyses of cytokines/chemokines profiles by comparing conditioned media prepared from TRAMP-C1, PBMC from normal C57BL/6 mice (PBMC#1, PBMC#2), and three different TILs (TILs #1-#3). Reduction of secreted proteins in the conditioned media are labeled with arrows. Induction of IL1 receptor antagonist (IL1RN) is enclosed with a dashed box. (**B**) Elevated expression of *Il1rn* mRNA in tumor tissues (tumor #1-#3). (**C**) Comparison of *Il1rn* mRNA expressions between tumor tissues (tumor#1-#3) and their derived TILs (TILs #1-#3). The samples are the same as described in Figure 3 and *Il1rn* levels are elevated in TILs. qPCR measurements were derived from three replicates and results are presented as the mean ± SD. Student’s *t*-test. * *p* < 0.05, *** *p* < 0.001, **** *p* < 0.0001.

**Figure 5 biology-09-00067-f005:**
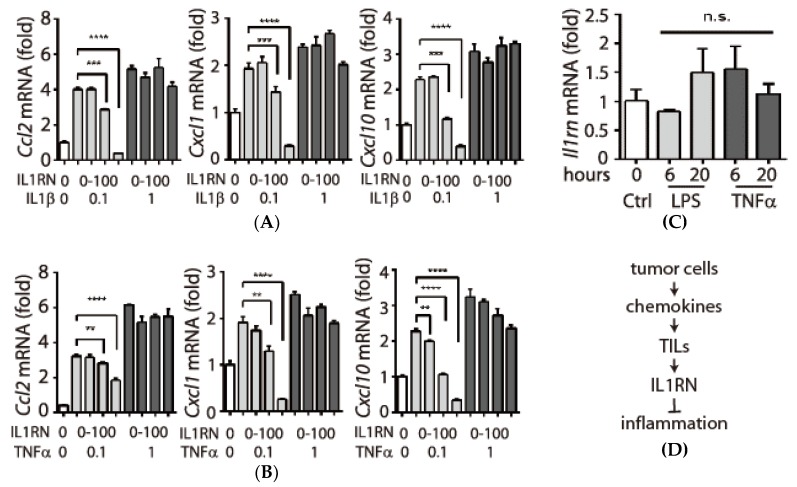
Inhibitory effects of interleukin-1 (IL1) receptor antagonist (IL1RN) to pro-inflammatory cytokines. (**A**) Monitoring inhibitory effects of IL1RN on IL1β. Different concentrations of IL1RN proteins (0, 1, 10, and 100 ng/mL, labeled as a triangle) were mixed with a fixed dosage of IL1β (0.1 or 1 ng/mL) in TRAMP-C1 cells for 6 h. Relative mRNA levels were measured by primers targeting *Ccl2* (left), *Cxcl1* (middle), and Cxcl10 (right). (**B**) Monitoring inhibitory effects of IL1RN on TNFα (0.1 or 1 ng/mL). (**C**) *Il1rn* mRNA was not affected by inflammatory signaling. TRAMP-C1 cells were treated with lipopolysaccharide (LPS, 100 ng/mL) or mouse tumor necrosis factor-α (TNFα, 10 ng/mL) for 6 or 20 h followed by mRNA purification and a real-time qPCR analysis. qPCR measurements were derived from three replicates and results are presented as the mean ± SD. Student’s *t*-test. ** *p* < 0.01, *** *p* < 0.001, **** *p* < 0.0001. (**D**) A working model depicts the recruitment of TILs by tumor-associated pro-inflammatory chemokines (e.g., CCL2). The recruited TILs secret IL1RN which suppresses inflammatory signals by IL1RN-mediated functions in tumor microenvironment.

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
