# Peer review of "Tumor-infiltrating Leukocytes Suppress Local Inflammation Via Interleukin-1 Receptor Antagonist in a Syngeneic Prostate Cancer Model"

_biology, 2020, doi:10.3390/biology9040067_

Round 1
Reviewer 1 Report
The authors have adequately addressed all points.
Author Response
Thanks again for the reviewer’s constructive comments. We will continue the research and utilize our resource more efficiently to work on the mechanism.
Reviewer 2 Report
The manuscript has been significantly improved and ready for publication.
Author Response

(The authors gave the same response as above.)

Reviewer 3 Report
The authors herein examined tumorigenic chemokines in a syngeneic prostate cancer model and found that tumor cells recruit tumor-infiltrating leukocytes (TILs) by pro-inflammatory chemokines to establish an interleukin-1 receptor antagonist (IL1RN)-mediated anti-inflammatory environment. The findings are interesting and could be useful for the future exploration to understand the tumor microenvironment, but several modifications are necessary to further clarify their argument.
1. As a contrasting experiment, it would be better to see the level of IL1RN in the tumor tissue of the Lewis lung cancer cell line.
2. Please clearly indicate what “primary cells” mean in the Result section since it is a rather confusing terminology.
3. They should check the types of infiltrated lymphocytes by the markers (e.g. B cell, T cell, CD4, CD8, etc).
4. It would be preferable to confirm which cells produce IL1RN among TILs.
5. The meaning of the following sentence is unclear: “Thus, the late stages and malignant tumors can escape the immune system by reducing the IL1RN level in the tumor microenvironment” (page 9, lines 280-281). How can tumor cells escape the immune system by reducing the IL1RN level, which should possibly induce much more inflammatory reaction.
6. Please fix the following: Supplementary Fig. S1C→Supplementary Fig. S1B (page 9).
Author Response
We first thank for the reviewer’s constructive comments. Please see the attachment.

Round 2
Reviewer 3 Report
The reviewer would like to thank the authors for having carefully and sufficiently addressed all the questions and comments raised here. The manuscript should be accepted without any further revision.